# Comprehensive Analysis of Differentially Expressed Unigenes under NaCl Stress in Flax (*Linum usitatissimum* L.) Using RNA-Seq

**DOI:** 10.3390/ijms20020369

**Published:** 2019-01-16

**Authors:** Jianzhong Wu, Qian Zhao, Guangwen Wu, Hongmei Yuan, Yanhua Ma, Hong Lin, Liyan Pan, Suiyan Li, Dequan Sun

**Affiliations:** 1Institute of Forage and Grassland Sciences, Heilongjiang Academy of Agricultural Sciences, Harbin 150086, China; wujianzhong176@163.com (J.W.); mayanhua@163.com (Y.M.); linhong@163.com (H.L.); panliyan@163.com (L.P.); lisuiyan@163.com (S.L.); 2Institute of Industrial Crop, Heilongjiang Academy of Agricultural Sciences, Harbin 150086, China; zhaoqian0401@sina.com (Q.Z.); wuguangwenflax@163.com (G.W.); yuanhm@163.com (H.Y.)

**Keywords:** RNA-seq, DEUs, flax, NaCl stress, EST-SSR

## Abstract

Flax (*Linum usitatissimum* L.) is an important industrial crop that is often cultivated on marginal lands, where salt stress negatively affects yield and quality. High-throughput RNA sequencing (RNA-seq) using the powerful Illumina platform was employed for transcript analysis and gene discovery to reveal flax response mechanisms to salt stress. After cDNA libraries were constructed from flax exposed to water (negative control) or salt (100 mM NaCl) for 12 h, 24 h or 48 h, transcription expression profiles and cDNA sequences representing expressed mRNA were obtained. A total of 431,808,502 clean reads were assembled to form 75,961 unigenes. After ruling out short-length and low-quality sequences, 33,774 differentially expressed unigenes (DEUs) were identified between salt-stressed and unstressed control (C) flax. Of these DEUs, 3669, 8882 and 21,223 unigenes were obtained from flax exposed to salt for 12 h (N1), 24 h (N2) and 48 h (N4), respectively. Gene function classification and pathway assignments of 2842 DEUs were obtained by comparing unigene sequences to information within public data repositories. qRT-PCR of selected DEUs was used to validate flax cDNA libraries generated for various durations of salt exposure. Based on transcriptome sequences, 1777 EST-SSRs were identified of which trinucleotide and dinucleotide repeat microsatellite motifs were most abundant. The flax DEUs and EST-SSRs identified here will serve as a powerful resource to better understand flax response mechanisms to salt exposure for development of more salt-tolerant varieties of flax.

## 1. Introduction

Worldwide, flax (*Linum usitatissimum* L.) is an economically important fiber crop, with the flax fiber industry rapidly expanding to meet increasing demand. However, in China flax competes with higher priority food crops, such as grains, that require ever increasing acreage to meet increasing food demands [1]. Consequently, flax cultivation there is confined to barren or even high-salinity plots, where flax varieties with high salt stress tolerance are urgently needed to increase yields of high quality fiber [2]. Unfortunately, the current lack of suitable salt tolerant varieties awaits development of salt-tolerant and high-yield flax germplasm resources. Toward this end, a current research focus is to identify flax genes involved in salt tolerance and salt stress responses [3]. To date, studies of flax salt stress responses have mainly focused on physiological and biochemical aspects instead of molecular response mechanisms. With the completion of the flax genome sequence [4] and continuous development of powerful molecular biological techniques, tools are now available to study regulatory responses to salt stress at the molecular level. Because the neutral salt NaCl is the main source of harmful salt in saline-alkali soils found on marginal lands, flax varieties that are specifically tolerant to NaCl are desired in China [5]. Only after a better understanding of complex salt tolerance mechanisms is mastered will great strides me made toward successful breeding of flax varieties with greater resistance to salt stress.

Research on mechanisms of salt tolerance in plants other than flax has developed rapidly and has revealed that specific molecular mechanisms of salt and alkali tolerance tend to be very complicated [6]. Plants employ adaptations and morphological changes to cope with various abiotic stresses through molecular, cellular, physiological and biochemical responses to stressors [7]. For example, plants regulate and balance osmotic pressure inside and outside of cells by accumulating metabolites to reduce or eliminate stress damage caused by water loss [8]. Nitric oxide (NO) was one of the factors which was operating the melatonin downstream to promote salinity tolerance in rapeseed based on the pharmacological, molecular and genetic data [9]. Arabidopsis EARLY FLOWERING3 (ELF3) enhances plants’ resilience to salt stress [10]. Under salt stress, ELF3 suppresses GIGANTEA (GI) at the post-translational level and PHYTOCHROME INTERACTING FACTOR4 (PIF4) at the transcriptional level and PIF4 directly up-regulates the transcription of ORESARA1 and SAG29, which were the two genes that are positive regulators of salt stress response pathways.

Effects of salt stress on plant morphological development are mainly observed as reductions in seed germination, seedling growth and altered growth and development of plant tissues and organs [11,12,13]. Under salt stress, plant growth is generally inhibited by a water deficit manifesting as water exosmosis from cells. Exosmosis decreases plant growth rate significantly, causing wilting of plants and cell membrane damage that leads to plant cell death [14,15,16]. Meanwhile, Na^+^ competition with various nutrients prevents plants from absorbing other key mineral elements, causing nutrient deficits such as K^+^ deficiency, the most common NaCl-induced nutrient deficiency observed [17,18]. So far, a large number of studies on salt tolerance have been carried out in *Arabidopsis thaliana* [19,20], Oryza sativa [21,22] and other crops [23,24,25,26,27,28] using RNA-Seq technology. The integration of spatiotemporal expression patterns and response characteristics of different genes helps to identify a large number of differentially expressed genes (DEGs) and mechanisms related to salt stress.

In recent years, salt damage has seriously affected flax production in northeast China. Therefore, it is of great significance to understand salt stress response mechanisms and signal pathways, a challenge which currently has important economic, environmental and scientific urgency. Exploring the functional molecules of salt stress signals is fundamental to understanding the mechanism of salt-tolerant crops, so as to conduct genetic engineering and accelerate the breeding of salt-tolerant crops [29]. Ultimately, future agricultural breeding programs will benefit from enhancement of our understanding of resistance mechanisms to a variety of other stressors as well [30].

Transcriptome analysis, a recently developed tool, has greatly enhanced our understanding of plant stress resistance mechanisms [21,23,25,26,27,31]. However, this powerful technology has seldom been applied to the study of molecular mechanisms involved in flax tolerance, with only a few known resistance genes characterized to date. Here, we sequenced the flax transcriptome to identify differentially expressed unigenes (DEUs) for different NaCl stress exposure durations to better understand flax adaptive molecular responses mechanisms under NaCl stress. After flax transcriptome results were confirmed using qRT-PCR, large-scale analysis of EST-SSRs was conducted using public resources to understand the functions of genes involved in salt stress. The information obtained from this work lays the foundation for understanding molecular mechanisms that participate in the flax response to salt and other stressors, while also identifying relevant and useful genes and markers for future development of salt-tolerant flax varieties.

## 2. Results and Discussion

### 2.1. Response of Flax to NaCl Stress

Membrane systems are primary sites of salt stress injury, where such damage causes changes in or loss of plasma membrane semiperme ability, leading to increased electrolyte extravasation [8]. Because field experimentation is difficult to control and time-consuming, in its place laboratory studies have been conducted that measure plant leaf electrical conductivity to study salt stress. Based on previous preliminary conductivity results, three exposure times (N1 for 12 h, N2 for 24 h or N4 for 48 h) were selected to measure changes in gene expression profiles of flax exposed to 100 mM NaCl solution for each exposure duration in the laboratory.

### 2.2. Transcriptome Analysis

Illumina paired-end sequencing technology was employed to explore DEUs related to NaCl stress in flax using two biological replicates per time point. A total of 185,457,832 clean reads were generated after removing low quality regions and adapter sequences and sequences were mapped to the flax genome (ftp://climb.genomics.cn/pub/10.5524/100001_101000/100081/Flax.cds) using bowtie 2 (v2.1.0) (https://sourceforge.net/projects/bowtie-bio/files/bowtie2/2.1.0/) for short reads with default parameter settings (Table 1). Only 42.92% of reads could be mapped to the reference genome, possibly due to incomplete flax genome assembly. After RSEM (V1.2.4) (https://omictools.com/rsem-tool) [32] was used to evaluate the expression level of each gene, 304,809 of 422,316 unigenes remained with at least 1 fragment per kilobase of transcript per million mapped reads (FPKM) (Table 2). Sequence regions were assigned to exon, intron and intergenic types after comparing total mapped sequence reads with the reference genome. Since exon-type sequences accounted for more than 80% of genome-mapped sequences, as expected (Figure 1), these results confirmed a high degree of annotation accuracy. Moreover, comparison of FPKM box plots of gene expression levels of all genes for different experimental conditions demonstrated that sequence results were reliable, since each sample yielded equivalent reads and coverage depths between duplicates (Figure 2). Furthermore, comparison rates between sequence data and reference genes also demonstrated sequence reliability to some extent.

### 2.3. Identification of Differentially Expressed Unigenes (DEUs)

Gene expression profiles in response to NaCl stress exposures of 12 h (N1), 24 h (N2) and 48 h (N4) were compared to the no-treatment control (water). DEUs were identified within the set of transcriptome sequences, with 33,774 significant DEUs identified (18,040 up-regulated and 15,734 down-regulated), with values of false discovery rate (FDR) ≤ 0.05 and 2× fold change significance cutoffs generated for various treatment time points.

At 12 h, 24 h and 48 h of stress exposure, 3669 DEUs (2219 up-regulated and 1450 down-regulated), 8882 DEUs (4865 up-regulated and 4017 down-regulated) and 21,223 DEUs (10,956 up-regulated and 10,267 down-regulated) were significantly differentially regulated in response to NaCl stress exposure, respectively (Figure 3). While different numbers of DEUs were observed for various pairings of stress exposure durations, 2581 DEUs (11.5%) were identified that were shared among all stress-exposed samples (Figure 4), in which 2576 co-expressed unigenes were detected (1322 up-regulated and 1254 down-regulated), with five unigenes not co-expressed (Appendix A). The proportion of DEUs common to paired exposure time points of 24 h/48 h (25.2%) was higher than corresponding proportions for 12 h/24 h (0.9%) and 12 h/48 h (1.9%). These results may be attributed to relatively greater effects of salt injury stress on flax from 24 h to 48 h than for other exposure windows, with involvement of a larger number of regulatory genes observed. This result is not surprising, since expression trends of common DEUs were not entirely consistent among different time periods. In general, DEUs analysis of flax under NaCl stress exposures should enhance our understanding of factors influencing gene expression during salt stress responses and provide clues to genes involved in salt tolerance.

### 2.4. Cluster Analysis of DEUs

Cluster analysis was used to determine expression patterns of DEUs under different experimental conditions. Generally, functions of unknown genes or unknown functions of known genes can be identified by clustering genes with the same or similar expression patterns into classes; genes with similar functions or genes that participate in the same metabolic processes or cell pathways tend to cluster together. DEU dynamic expression patterns for various NaCl stress exposure durations were identified in this study (Figure 5). After hierarchical clustering analysis of DEUs was conducted based on gene expression levels determined from FPKM values, DEUs within each single cluster were considered to be co-expressed genes. Color-coding of different cluster groupings highlights genes with similar expression patterns that shared similar functions or participated in the same biological processes. Moreover, duplicate biological samples for control or each NaCl stress treatment group were highly consistent, thus demonstrating reproducibility of RNA-seq results.

### 2.5. Functional Annotation

To understand DEU functions, we conducted Pfam, GO, KOG and KEGG enrichment analyses against the genetic background of the *Arabidopsis thaliana* genome (https://www.arabidopsis.org/). Of 2582 co-expressed unigenes, 2482 (96.13%) displayed significant similarity to known proteins (Appendix A), with 2104, 1473, 785 and 913 unigenes matched to homologous sequences using Pfam, GO, KOG and KEGG analyses, respectively. The most common enrichment analysis terms were related to pathway factors (Figure 6) such as plant hormone signal transduction, photosynthesis-antenna proteins and biosynthesis of amino acids as important in flax responses to NaCl exposure. Not surprisingly, here we identified a number of differentially expressed unigenes (DEUs), which were homologous to known stress regulating plant transcription factors, such as: bZIP (lus10033630), HD-ZIP (lus10025232), WRKY (Lus10020023, Lus10024380, Lus10022736, Lus10012870 and Lus10030517), NAC (Lus10042731, Lus10026617, Lus10036773, Lus10025118, Lus10041534, Lus10003269 and Lus10042518), MYB (Lus10006740, Lus10019085 and Lus10006647), GRF (Lus10015651 and Lus10037668), GATA (Lus10031464, Lus10025829 and Lus10038273), ERF (Lus10005285, Lus10029987 and Lus10012226), CAMTA (Lus10024044) and B3 (Lus10018583 and Lus10039816). The basic leucine-region zipper (bZIP) transcription factors (TFs) act as crucial regulators in salt stress responses in plants [33]. In our study, Lus10033630 was homologous to AtbZIP34 which was required for Arabidopsis pollen wall patterning and the control of lipid metabolism and/or cellular transport in developing pollen.

Interestingly, two salt-tolerant genes (lus10015754 and lus10000310) were obtained here to homologous with Arabidopsis Senescence-Associated Gene29 (SAG29), which was consistent with previous reports [10] and two salt-tolerant genes (lus10015285 and lus10025409) were homologous with SAG12 whose promoter control expression of isopentenyl transferase (ipt) gene in the decaying leaves of the lower part of the plant. Three genes (*Lus10040248*, *Lus10019786* and *Lus10037551*) belong to the Rho-like GTPases from plants (*ROP*) gene family, which might enhance salt tolerance by increasing root length, improving membrane injury and ion distribution [34]. Overall, all co-expressed unigenes could be aligned to the reference genome, suggesting that annotation and classification analyses performed here could be used to reliably predict flax gene functions.

### 2.6. RNA-Seq Expression Validation

To quantitatively assess the reliability of transcriptome data, six candidate DEUs were selected for analysis using real-time reverse transcription quantitative PCR (qRT-PCR) of biological duplicate samples. Consistent with RNA-sequencing analysis results, qRT-PCR showed significant log2-fold expression changes of DEUs among different salt-stress exposure treatments (Figure 7), thus demonstrating the reliability and accuracy of the transcriptome analysis of NaCl stress in flax conducted here.

### 2.7. Distribution Characteristics of EST-SSRs

A total of 1777 EST-SSRs with 2–6 bp repeat numbers were identified from sequence data (Table 3). Among SSR loci, trinucleotide microsatellites were the most abundant repeat motif (1002 SSRs, accounting for 56.39%), followed by dinucleotide microsatellites (623 SSRs, accounting for 35.06%), tetranucleotide microsatellites (68 SSRs, accounting for 3.83%), pentanucleotide microsatellites (58 SSRs, accounting for 3.26%) and hexanucleotide microsatellites (26 SSRs, accounting for 1.46%). A repeat iteration number of five (621) was the most common repeat iteration number observed, followed by six (502) and seven (258) repeat iteration numbers. AT/TA motifs were the dinucleotide microsatellite motifs most frequently observed of the six possible motifs (AC/TG, AG/TC, AT/TA, CT/GA, CG/GC and GT/CA), while CTT/GAA motifs were the most frequently represented trinucleotide microsatellites of the 30 possible motifs. The details of all EST-SSRs with the primer pairs are shown in Appendix A.

## 3. Materials and Methods

### 3.1. Material Planting and Processing

The fiber flax variety (Agatha), provided by Heilongjiang Academy of Agricultural Sciences in Harbin, China, provided material for high-throughput RNA-seq. Flaxseeds were placed in cups filled with sterilized vermiculite and maintained at 28 °C during the day and 22 °C at night in a growing room on a 16-h light/8-h dark cycle. Plants were irrigated every three days and the humidity in the growth room was maintained at 70%.

### 3.2. Stress Treatments and Sample Preparation

Three-week-old seedlings were rinsed to remove vermiculite and were placed into tanks filled with 100 mM NaCl solution or water for the control (CK). After seedlings were exposed to NaCl solution or water for 12 h, 24 h or 48 h, whole seedlings were harvested, frozen immediately in liquid nitrogen, then stored at −80 °C until RNA was prepared. Two biological replicates per treatment (each sample containing 10 plants each) were processed in parallel.

### 3.3. RNA Isolation and cDNA Library Construction

RNA isolation and cDNA library construction methods have been reported previously [35].

### 3.4. Illumina Sequencing, Assembly and Annotation

Transcriptome sequencing was performed using the Illumina HiSeq 2500 platform (https://www.illumina.com/systems/sequencing-platforms/hiseq-2500.html) to generate paired-end (PE) raw reads, each of ~100 bp in length. Clean reads were generated from raw reads by removal of adaptor sequences, ambiguous ‘N’ nucleotides (with a ratio of ‘N’ greater than 10%) and low quality sequences (with quality scores less than 5) and were assembled as described by using bowtie2 (2.1.0) software (http://bowtie-bio.sourceforge.net/bowtie2/index.shtml) against a reference genome (ftp://climb.genomics.cn/pub/10.5524/100001_101000/100081/). Only clean, high quality sequence data was used in subsequent analyses.

For homology-based annotation, the model plant genome of *Arabidopsis thaliana* (https://www.arabidopsis.org/) was selected for use as genetic background, with non-redundant sequences subjected to Pfam (http://pfam.xfam.org/), Gene Ontology (GO) (http://www.geneontology.org/), Eukaryotic Clusters of Orthologous Groups (KOG) (https://www.ncbi.nlm.nih.gov/COG/) and Kyoto Encyclopedia of Genes and Genomes (KEGG) (http://www.genome.jp/kegg/). For gene expression profiling analysis, functional assignments were mapped to GO terms [36]. Significantly enriched pathways were identified according to *p* values and enrichment factors [37].

### 3.5. Identification of DEUs and Cluster Analysis

DEUs were identified based on a negative binomial distribution using the edgeR package (https://www.rdocumentation.org/packages/edgeR/versions/3.14.0) [38]. Candidate genes exhibited false discovery rate (FDR) values ≤ 0.01, which were calculated from numbers of mapped reads. In addition, the fragments per kilobase of transcript per million mapped reads (FPKM) values of candidate genes were calculated using RSEM (http://deweylab.github.io/RSEM/; RNA-Seq by Expectation-Maximization) [32]. All expressed genes were divided into six categories: 0 < FPKM ≤ 0.1, 0.1 < FPKM ≤ 1, 1 < FPKM ≤ 3, 3 < FPKM ≤ 15, 15 < FPKM ≤ 60 and FPKM > 60. The formula for calculating expression level reflected by FPKM for each target gene is:
FPKM = 10C^9^/NL
where C is the number of fragments of the target gene, N is the number of fragments for all genes and L is the base length of the target gene.

Finally, the fold change of FPKM for each sequence was calculated and genes with FPKM fold changes greater or equal to 2 were classified as DEUs. Common DEUs among different time points that exhibited FPKM fold change differences greater than 2 between biological replicates were eliminated. For each gene, normalized FPKM values for each transcript were clustered using the hclust function in R (http://www.r-project.org/) using a distance matrix representing FPKM level profiles of genes across the four sampled time points. The tree produced by the clustering process was split into two branches using the cutree function.

### 3.6. Real-Time qRT-PCR Validation

DEUs identified in transcriptome sequencing analysis were validated using qRT-PCR (quantitative RT-PCR) to further confirm RNA-Seq analysis gene expression results. Six genes (*Lus10017947*, *Lus10001671*, *Lus10009502*, *Lus10039893*, *Lus10034541* and *Lus10014048*) were selected for analysis of expression levels in flax treated with NaCl solution exposure times of 12 h, 24 h and 48 h using *L. ussitatissimum Act1* (GenBank accession no. AY857865) as the internal reference gene. Primer sets were designed from Illumina sequencing data using Primer Premier v6.24 (http://www.premierbiosoft.com/crm/jsp/com/pbi/crm/clientside/ProductList.jsp) (listed in Appendix A). Primers were synthesized by GENEWIZ, Inc. (Suzhou, China). Quantitative RT-PCR was performed with a SYBR^®^ Premix Ex Taq™ II Kit (TaKaRa, Dalian, China) using an ABI 7500 Real-Time PCR System (Applied Biosystems, Foster City, CA, USA). Data for each experimental sample was corrected for sample loading differences using results of flax *Act1* qRT-PCR using the 2^−ΔΔCt^ method [39]. PCR amplification was performed using thermal cycling conditions of denaturation at 95 °C for 30 s followed by 40 cycles of amplification (95 °C for 5 s, 60 °C for 34 s). The control reaction (to normalize expression levels) and all samples were tested in triplicate.

### 3.7. Development of EST-SSR

All DEUs identified by transcriptome sequencing were analyzed to identify SSRs with dimer, trimer, tetramer, pentamer and hexamer motifs using SSRIT (http://archive.gramene.org/db/markers/ssrtool). SSR primer pairs were designed with highest primer scores using Primer-BLAST (https://www.ncbi.nlm.nih.gov/tools/primer-blast/index.cgi?LINK_LOC=BlastHomeAd) with the following standard parameters: target amplicon length of 100–300 bp, annealing temperature variation from 55 to 65 °C and primer size of 18–28 bp.

## 4. Conclusions

As an important industrial crop, flax is currently cultivated on marginal lands in the northeast region of China. By analyzing the genome-wide transcriptome of flax during exposure to 100 mM NaCl solution, we identified 3669, 8882 and 21,223 potential salt stress-responsive DEUs after salt exposure for 12 h, 24 h and 48 h, respectively, compared with untreated control. All of these unigenes were identified as part of an extensive investigation of flax genes involved in the response to salt exposure to collectively provide high-resolution gene expression profiles for flax under control and salt stress conditions. Of the 2582 co-expressed unigenes, 2482 were annotated using at least one public database (GO, KOG, KO, Pfam, KEGG) and pathways linked to flax response to NaCl exposure were mainly associated with plant hormone signal transduction, photosynthesis-antenna proteins and biosynthesis of amino acids. Number of the genes are homologous to known stress regulators. Transcriptome sequencing results were verified by qRT-PCR. Ultimately, 1777 EST-SSRs based on transcriptome results were identified as important in flax response to NaCl exposure. The results described here thus lay the groundwork for further elucidation of molecular mechanisms of flax stress responses. Ultimately, this information will guide the development of flax varieties that are more tolerant to a wide range of environmental stresses.

## Figures and Tables

**Figure 1 ijms-20-00369-f001:**
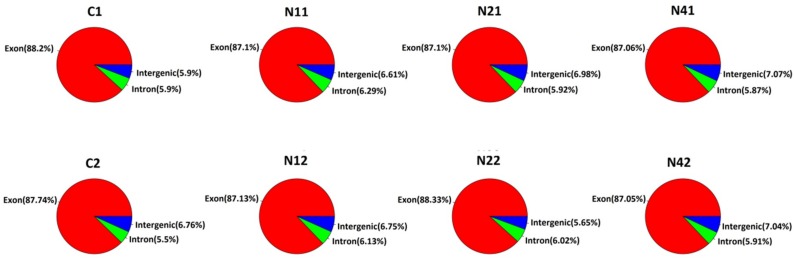
The comparison of clean reads with reference genome in different regions.

**Figure 2 ijms-20-00369-f002:**
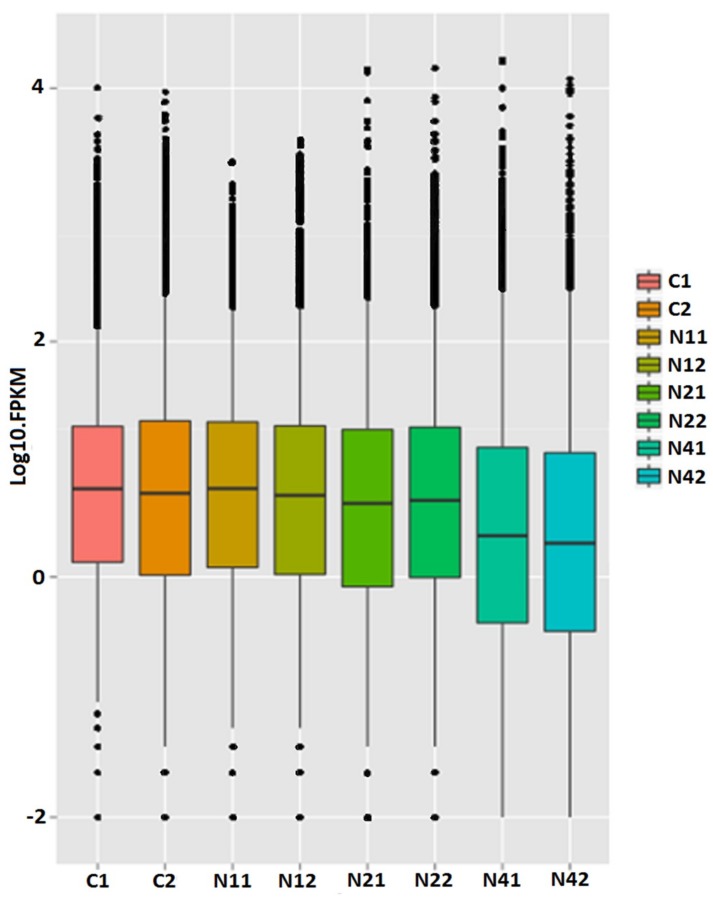
The box plot of FPKM in gene expression levels. Abscissa for sample name, ordinate for Log10.FPKM, box plot for each region for five statistics (From top to bottom: maximum, upper quartile, median, lower quartile and minimum), the outlier is shown in black dots.

**Figure 3 ijms-20-00369-f003:**
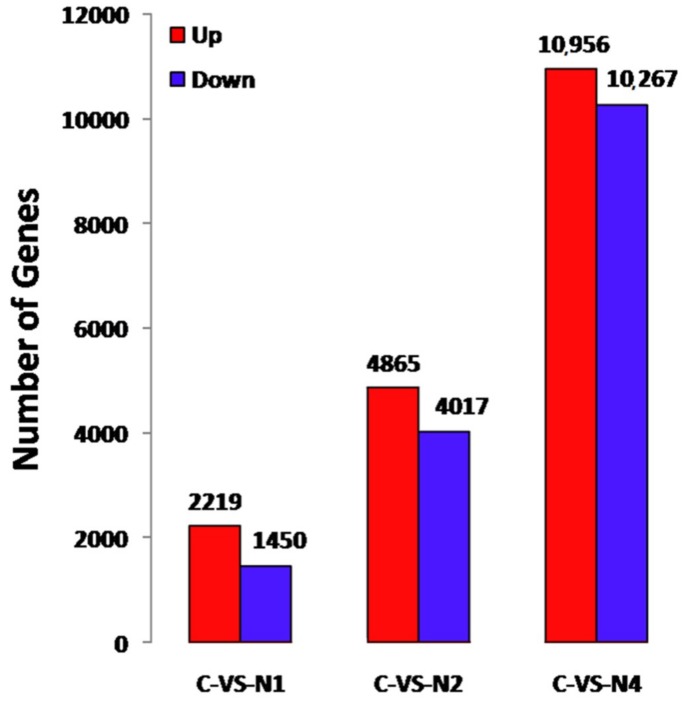
Comparison of up and down-regulation of DEGs. C-vs-N1, C-vs-N2 and C-vs-N4 representing the DEUs under the exposure time of 12 h, 24 h and 48 h in NaCl solution, respectively.

**Figure 4 ijms-20-00369-f004:**
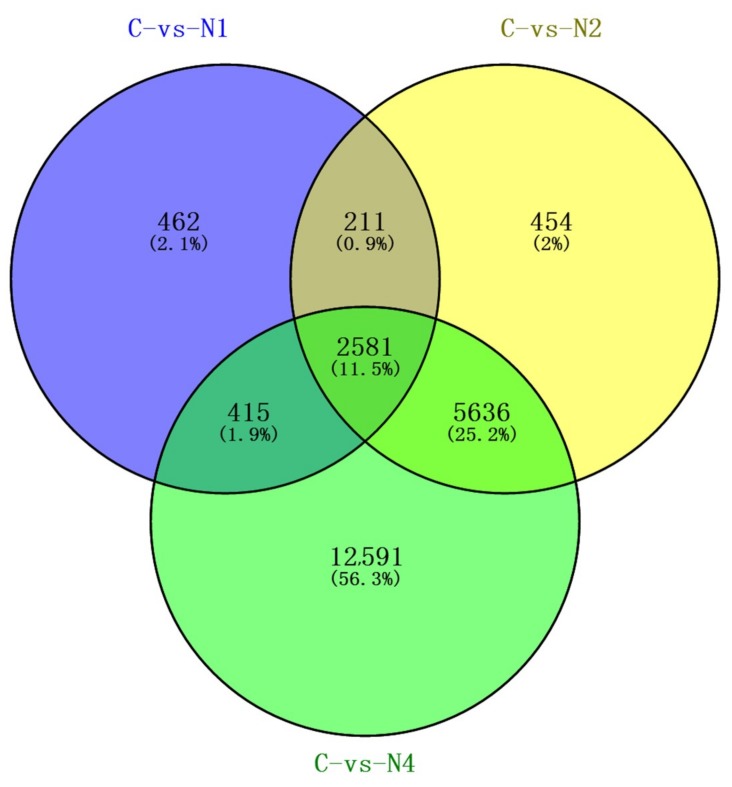
The Venn diagram of DEUs. Venn diagram representing the distribution of NaCl-responsive genes. The numbers in the Venn diagram indicated total numbers of regulated genes in the unique treatment.

**Figure 5 ijms-20-00369-f005:**
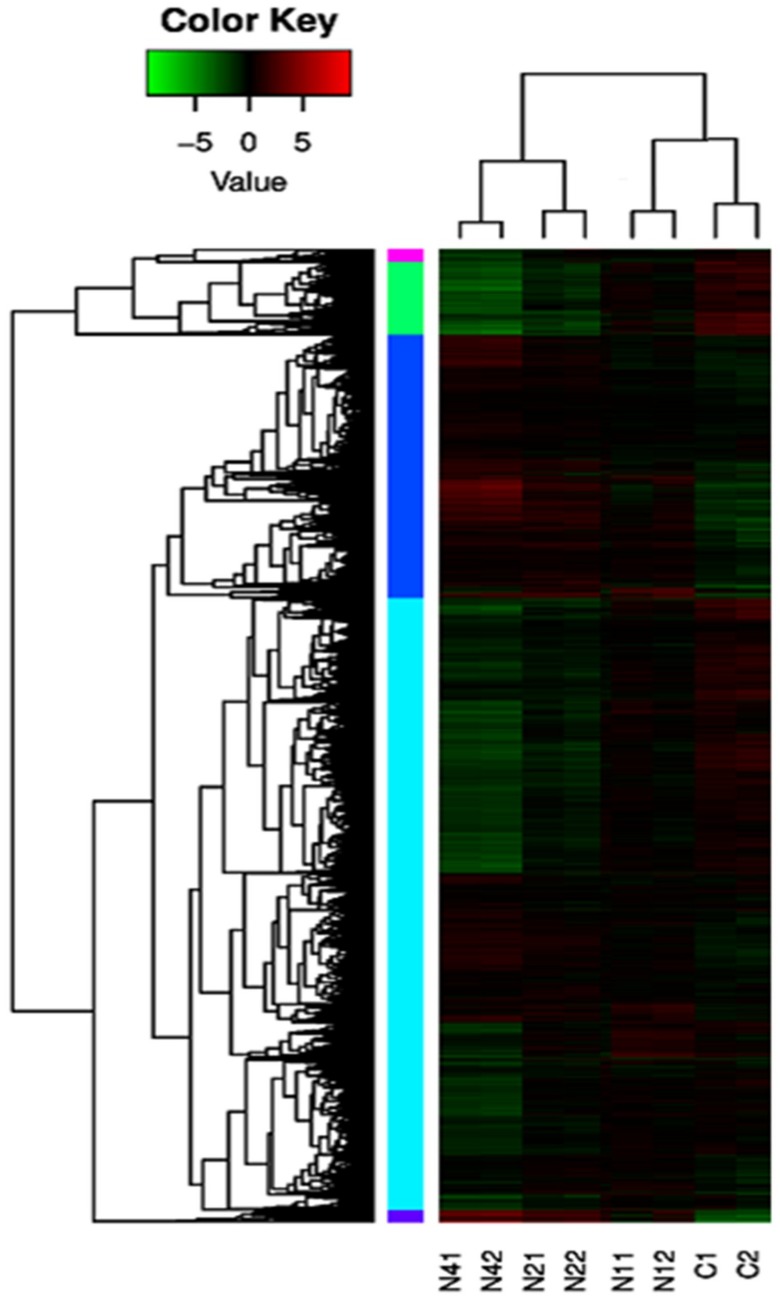
Cluster diagram of DEUs (FDR ≤ 0.05, fold change ≥ 2). The darker color represents the higher of the gene expression level. Each color block on the left represents a cluster of genes with similar expression levels. The Log2.FPKM value was used for clustering, with red for high expression gene and green for low expression gene. The color ranges from green to red, indicating higher gene expression.

**Figure 6 ijms-20-00369-f006:**
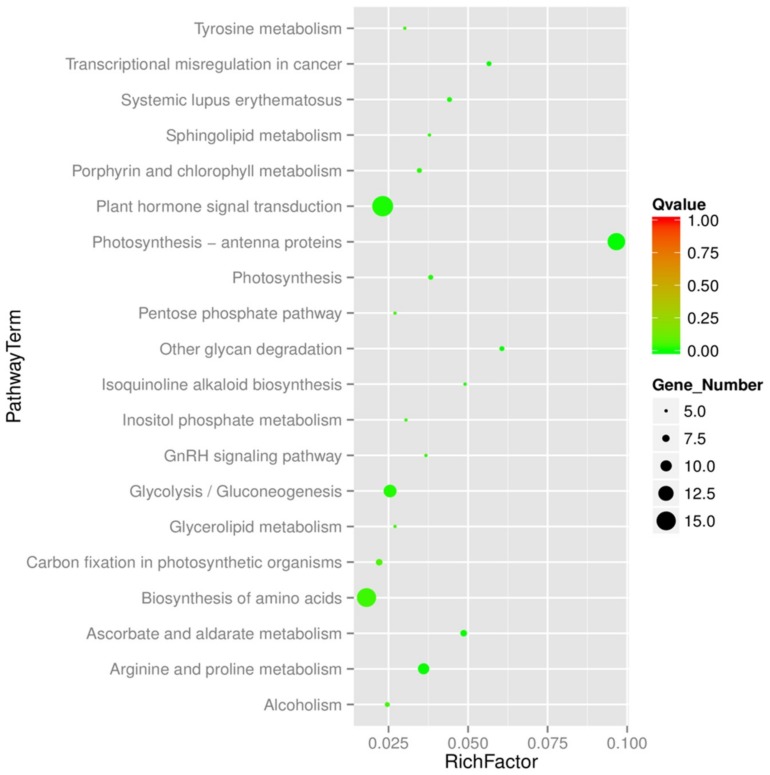
Gene enrichment of non-redundant unigenes. The longitudinal axis represents the different pathway and horizontal axis represents the Rich factor. The size of dots indicates the number of differentially expressed genes in this pathway and the color of the point corresponds to a different Qvalue range.

**Figure 7 ijms-20-00369-f007:**
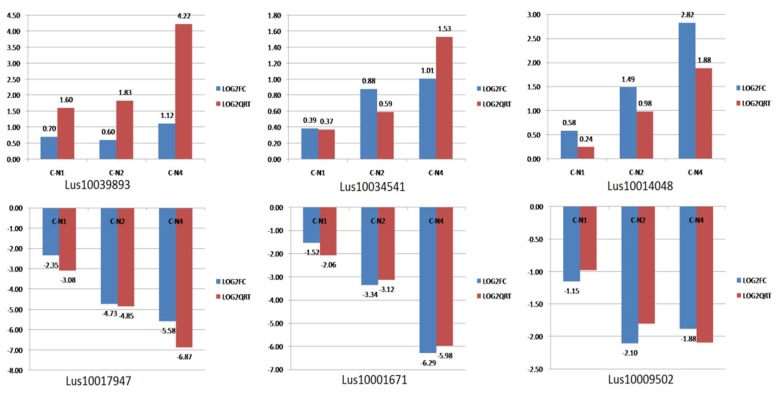
Validation of the RNA-seq data expression profile by qRT-PCR. The relative expression levels of 6 DEUs were calculated according to the 2^−ΔΔCt^ method using the *Actin* gene as an internal reference gene. The *x*-axis indicates the different exposure of 100 mM NaCl solution with 12 h (C-N1), 24 h (C-N2) and 48 h (C-N4). LOG2FC and LOG2QRT represent for the binary logarithm of fold changes of differentially expressed genes in RNA-seq and qRT-PCR, respectively.

**Table 1 ijms-20-00369-t001:** Summary of data generated in the transcriptome sequencing of flax.

Sample *	Clean Reads	Total Mapped	Unique Mapped	Multi Mapped	Bases (Gb)	Q20 (%)	GC (%)
C1	60,125,334	25,807,774	9,037,386	16,770,388	7.48	99.36	48.35
C2	54,128,778	23,792,226	9,600,424	14,191,802	6.67	99.23	47.68
N11	62,422,020	26,632,880	9,637,906	16,994,974	7.77	99.33	47.67
N12	57,381,976	22,307,914	8,055,084	14,252,830	7.13	99.3	47.84
N21	60,066,230	25,957,230	9,698,022	16,259,208	7.48	99.37	47.87
N22	21,827,146	9,277,362	3,136,434	6,140,928	2.58	97.64	47.81
N41	56,385,726	25,227,542	9,202,726	16,024,816	7.01	99.35	47.96
N42	59,471,292	26,454,904	9,667,984	16,786,920	7.39	99.31	48.03

* Samples: Name of sequencing sample, C1/C2 for the two biological replicates of control, N11/N12, N21/N22 and N41/N42 represent for the two biological replicates of treatment with 100 mM NaCl for 12 h, 24 h and 48 h respectively; Clear reads: Number of clean reads participating in the comparison; Total mapped: Numbers of all reads compared to reference genes; Unique mapped: The number of reads compared to the unique location of the reference gene which was used for gene expression analysis; Multi mapped: Number of reads compared to multiple locations of reference genes; Bases (Gb): The total number of bases, Gb represent for billion base pairs; Q20: Percentage of sequencing error rate is smaller than 1% of base number; GC(%):The percentage of G + C of the total number bases.

**Table 2 ijms-20-00369-t002:** The number of genes in different expression levels.

FPKM Interval	0–0.1	0.1–1	1–3	3–15	15–60	>60
C1	1635 (3.03%)	9893 (18.31%)	11,852 (21.93%)	21,176 (39.19%)	7660 (14.17%)	1823 (3.37%)
C2	1805 (3.37%)	11,354 (21.21%)	11,228 (20.97%)	18,620 (34.78%)	8258 (15.42%)	2278 (4.25%)
N11	1805 (3.31%)	10,645 (19.52%)	11,210 (20.56%)	20,619 (37.82%)	8343 (15.30%)	1901 (3.49%)
N12	1784 (3.28%)	11,534 (21.18%)	11,789 (21.65%)	19,438 (35.70%)	7855 (14.43%)	2047 (3.76%)
N21	2226 (4.15%)	12,475 (23.27%)	11,564 (21.57%)	17,954 (33.49%)	7324 (13.66%)	2061 (3.84%)
N22	1015 (2.03%)	11,621 (23.24%)	11,373 (22.75%)	16,902 (33.80%)	7049 (14.10%)	2039 (4.08%)
N41	3221 (6.32%)	15,996 (31.38%)	10,872 (21.33%)	13,032 (25.57%)	5696 (11.17%)	2154 (4.23%)
N42	3804 (7.43%)	16,694 (32.61%)	10,687 (20.88%)	12,463 (24.35%)	5431 (10.61%)	2111 (4.12%)

**Table 3 ijms-20-00369-t003:** Distribution of EST-SSR types.

Type	Repeat Number	Percentage (%)
4	5	6	7	8	9	10	>10	Total
**Dinucleotide**			238	148	82	57	38	60	623	35.06
**Trinucleotide**		558	250	107	49	36	2		1002	56.39
**Quadnucleotide**		53	12	3					68	3.83
**Pentanucleotide**	49	7	2						58	3.26
**Hexanucleotide**	23	3							26	1.46
**Total**	72	621	502	258	131	93	40	60	1777

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
