# Peer review of "Comprehensive Analysis of Differentially Expressed Unigenes under NaCl Stress in Flax (*Linum usitatissimum* L.) Using RNA-Seq"

_ijms, 2019, doi:10.3390/ijms20020369_

Round 1

Reviewer 1 Report

Dear Authors,

Reviewer comments ijms-410872

The manuscript entitled „Comprehensive analysis of differential expression unigenes under NaCl stress in flax (Linus ussitatissimum L.) using RNA-seq“ represents a useful study aimed at an investigation of NaCl-responsive transcripts in flax seedlings subjected to a short-term salinity stress (100 mM NaCl for 12, 24 and 48 h). The experimental data are presented appropriately, RNA-seq analysis results were validated by qRT-PCR of selected transcripts. The presented data are appropriately interepreted and the methods are adequately described.

I have only a few formal comments on the manuscript.

In Results and Discussion, Table 1, the samples description and marking have to be explained appropriately, i.e., C (C1, C2) for control samples, N1 (N11, N12) for 12 h NaCl treatment, N2 (N21, N22) for 24 h NaCl treatment, and N4 (N41, N42) for 48 h NaCl treatment.

Terminology:

Lines 165, 282: Correct the term „photosynthesis-antenna proteins“ (not „photosynthese-antenna proteins“).

Conclusions, line 279: Use the term „control conditions“ instead of „normal conditions“ as an opposite to salt treatments.

Further formal comments:

Abstract, line 17; Results and Discussion, line 86: Add a space between „100 mM“ and „NaCl“.

Abstract, line 22: Add a space between the words „2,842“ and „unigenes.“

Results and Discussion, line 107: Replace the word „in“ by „of“ in Table 2 legend, i.e., „Table 2. The number of genes of different expression levels.“

Results and Discussion, line 126: Correct the typing error in the word „identified“ (not „indentified“).

Figure 7 legend, line 186: Add a space between the words „100 mM NaCl solution“.

Experimental section, line 209: Correct the typing error in „100 mM NaCl solution“ (not „100Mm NaCl solution“).

Experimental section, line 244: Remove the word „of“ in the sentence „…in which the FPKM value differences greater than 2-fold were eliminated between the biological replicates.“

Experimental section, line 250: Add a space between the words „qRT-PCR“ and „validation“ in the heading „3.6. Real-time qRT-PCR validation“.

Conclusions, line 279: Use the term „control conditions“ instead of „normal conditions“ as an opposite to salt treatments.  Conclusions, line 282: Correct the term „photosynthesis-antenna proteins“ (not „photosynthese-antenna proteins“).

Final recommendation: Accept after a minor revision.

Author Response

Dear Reviewer,

Thank you for your comments and the recognition on manuscript entitled “Comprehensive analysis of differentially expressed unigenes under NaCl stress in flax (Linus ussitatissimum L.) using RNA-seq”.

  We have modified the manuscript as your suggestion.

1. The note of explanations was added in Table 1;

2. The term “photosynthesis-antenna proteins” was corrected;

3. The term “normal conditions” was replaced as “control conditions” in line 279 of the Conclusions section;

4. Add a space between the necessary words;

5. We revised the words and terms throughout the full text;

The revised manuscript will be re-uploaded soon!

Merry Christmas and thank you again!

Yours,

Wu Jianzhong

Reviewer 2 Report

In this manuscript, the authors exposed Flax plants to salt stress and performed RNA-Seq analysis to identify differentially expressed unigenes as a first step towards understanding the flax salt stress responses. They identified 3,669, 8,882, and 21,223 DEUs after exposure to 12 h, 24h and 48h NaCl stress, respectively. The results generated in this manuscript represent valuable information and could be used to better understand the flax responses to salt stress.

General comments:

·         The manuscript needs to be rewritten by a native English speaker as it contains hundreds of grammatical mistakes and sentences that are not clear.

·         The way it reads now, this is a descriptive manuscript where the authors only present the data as is with minimal explanation. They need to explain what information the data provides in terms of the flax response to salt stress.

·         Some of the analysis conducted in the manuscript should be better explained.

Specific comments:

·         The authors did not mine the data properly and only superficially discussed the major findings. Thus, the presentation of the results needs to be improved. For example, the authors performed GO enrichment analysis but did not discuss in detail their findings. They need to delve in and look at the genes within each enriched category and compare their results with stress responses of other plant species such as Arabidopsis, tomato, rice, etc. For the wealth of valuable expression data they generated, their only conclusion was “In terms of the enrichment analysis of related pathway factors (Figure 6), plant hormone signal transduction, photosynthese-antenna proteins and biosynthesis of amino acids were the most important pathway in NaCl tolerance to flax”.

·         I could not access the supplemental materials because the zipped file wouldn’t unzip and the provided link on the manuscript is broken. So I cannot comment on the supplemental material.

·         It is surprising to me that only 44% of the reads mapped to the reference genome. I have done RNA-Seq analysis with Arabidopsis, tomato, sunflower, and bean and have 75-98% of my reads mapping to the reference genome. The authors need to explain the low percentage of mapped reads.

·         In the materials and methods section, the authors mention “Differentially expressed unigenes (DEUs) were identified based on the negative binomial distribution with the edgeR package”. However in the Results and Discussion section, the authors mention “Bioconductor package DESeq (V1.14.0) [20] was used to identify the differentially expressed unigenes (DEUs) in transcriptome sequences”. The authors need to rewrite the materials and methods section.

·         Similarly in figure 4, the authors show that 11.5% of the DEUs are common among the three timepoints, 0.9% between 12h and 24h, 25.2% between 24h and 48h, and 1.9% between 12h and 48h. These results need to be discussed and explained. Why so many common DEUs between 24h and 48h and not in any of the other comparisons and what does it mean? What are the categories of genes represented in the common and unique genes in each comparison and what conclusions can you make based on these results regarding the flax response to salt stress? Instead the authors only say “That’s not surprising that the expression trend of common DEUs were not entirely consistent in different periods” which doesn’t give any information.

·         The figure one is not informative and need to be replaced since the authors do not further investigate the reads that mapped to introns and intergenic regions of the reference genome. Reads mapping to regions outside exons suggest inaccurate that the annotation of the reference genome. But it is not relevant to this manuscript unless the authors use this information to improve the genome annotation.

·         Same with table 1. Not informative. I would rather use the space to present another information.

·         In the abstract the authors say “High-throughput RNA sequencing (RNA-seq) technology provides a powerful and efficient method for transcript analysis and resistant gene discovery by which we conducted to understand the mechanisms of salt tolerance. The last part of this sentence should be changed to “to understand the possible mechanisms of flax response to salt”. The data presented here does not investigate the mechanisms of salt tolerance because the authors did not compare salt tolerant and salt sensitive flax varieties. 

Some examples of grammatical mistakes

·         The title should read:

Comprehensive Analysis of Differentially Expressed Unigenes under NaCl Stress in Flax

(Linum usitatissimum L.) using RNA-Seq

·         Lines 12 and 13. The sentence is completely wrong. It is not clear to the reader what the authors are trying to say by “Flax is an important industrial crop and has been seriously attacked by salt stress, which requires available information for resistant genes on genome background”.

Author Response

Dear Reviewer,

Thank you for your comments and the recognition on manuscript entitled “Comprehensive analysis of differentially expressed unigenes under NaCl stress in flax (Linus ussitatissimum L.) using RNA-seq”. It is very kind of you to give the valuable comments.

We have modified the manuscript as your suggestion.

1. Professionals were invited to rewrite the manuscript to eliminate the grammatical mistakes and sentences that are not clear;

2. The supplemental materials have been re-uploaded for your valuable comments;

3. Table 1 was replaced with filling the information of Bases, Q20(%) and GC(%);

4. The abstract has been optimized;

5. The figure 1 was placed in the manuscript to better illustrate the accuracy of annotation against reference genome;

6. We have corrected the other questions you raised in the comments;

7. Only 42.92% of the reads mapped to the reference genome, this may be due to the incomplete assembly of the flax genome.

8. The common DEUs between 24h/48h (25.2%) was the most proportion than the DEUs between 12h/24h (0.9%), and 12h/48h (1.9%) time points. These may be attributed to that a large number of regulatory genes were expressed in flax salt injury stress within 24 to 48 hours.

Here, we identified large scale of DEUs after exposure to NaCl stress. The results could be used to better understand the flax responses to salt stress. In addition, we have listed the database annotation results of all unigenes and homologous annotation of related genes with Arabidopsis thaliana in Additional Table 2, which can be used as a supplement for enrichment analysis of genes.

The revised manuscript will be re-uploaded in this Friday.

Merry Christmas!

Thank you again for your careful review!

Yours,

Wu Jianzhong

Round 2

Reviewer 2 Report

.

Author Response

Response to Reviewer 2 Comments

Dear Reviewer 2,

Thank you for your comments and the recognition on manuscript.

I am very sorry for giving you not so good impression in the Round 1 process of manuscript modification. We have modified the manuscript as your suggestion.

Point 1: The manuscript needs to be rewritten by a native English speaker as it contains hundreds of grammatical mistakes and sentences that are not clear.

Response 1: Professionals were invited to rewrite the manuscript to eliminate the grammatical mistakes and sentences that are not clear.

Point 2: The authors did not mine the data properly and only superficially discussed the major findings. Thus, the presentation of the results needs to be improved.

Response 2: According to your suggestions, in the Introduction section, some cases of salt-tolerance gene reports have been added, such as NO, ELF3 and SAG29. Meanwhile, the research reports on salt tolerance in Arabidopsis thaliana, Oryza sativa and other crops were exemplified, and the related references were listed; In the Results and Discussion section, numbers of differentially expressed unigenes (DEUs), which were homologous to known stress regulating plant transcription factors, were added to discuss in detail the findings here. Emphatically, SAG29, SAG12 and ROP gene family were mentioned which have been recognized to play an important role in salt stress.

Point 3: I could not access the supplemental materials because the zipped file wouldn’t unzip and the provided link on the manuscript is broken. So I cannot comment on the supplemental material.

Response 3: The supplemental materials have been re-uploaded for your valuable comments.

Point 4: It is surprising to me that only 44% of the reads mapped to the reference genome. I have done RNA-Seq analysis with Arabidopsis, tomato, sunflower, and bean and have 75-98% of my reads mapping to the reference genome. The authors need to explain the low percentage of mapped reads.

Response 4: Only 42.92% of the reads mapped to the reference genome, this may be due to the incomplete assembly of the flax genome. We have mentioned this point in the revised manuscript.

Point 5: In the materials and methods section, the authors mention “Differentially expressed unigenes (DEUs) were identified based on the negative binomial distribution with the edgeR package”. However in the Results and Discussion section, the authors mention “Bioconductor package DESeq (V1.14.0) [20] was used to identify the differentially expressed unigenes (DEUs) in transcriptome sequences”. The authors need to rewrite the materials and methods section.

Response 5: The content was corrected, and we have rewritten the materials and methods section;

Point 6: Similarly in figure 4, the authors show that 11.5% of the DEUs are common among the three timepoints, 0.9% between 12h and 24h, 25.2% between 24h and 48h, and 1.9% between 12h and 48h. These results need to be discussed and explained. Why so many common DEUs between 24h and 48h and not in any of the other comparisons and what does it mean? What are the categories of genes represented in the common and unique genes in each comparison and what conclusions can you make based on these results regarding the flax response to salt stress? Instead the authors only say “That’s not surprising that the expression trend of common DEUs were not entirely consistent in different periods” which doesn’t give any information.

Response 6: The common DEUs between 24h/48h (25.2%) was the most proportion than the DEUs between 12h/24h (0.9%), and 12h/48h (1.9%) time points. These may be attributed to that a large number of regulatory genes were expressed in flax salt injury stress within 24 to 48 hours. We have written this inference into the manuscript.

Point 7: The figure one is not informative and need to be replaced since the authors do not further investigate the reads that mapped to introns and intergenic regions of the reference genome. Reads mapping to regions outside exons suggest inaccurate that the annotation of the reference genome. But it is not relevant to this manuscript unless the authors use this information to improve the genome annotation.

Response 7: Figure 1 was placed in the manuscript to better illustrate the accuracy of annotation against reference genome.

Point 8: Same with table 1. Not informative. I would rather use the space to present another information.

Response 8: Table 1 was replaced with filling the information of Bases, Q20(%) and GC(%).

Point 9: In the abstract the authors say “High-throughput RNA sequencing (RNA-seq) technology provides a powerful and efficient method for transcript analysis and resistant gene discovery by which we conducted to understand the mechanisms of salt tolerance. The last part of this sentence should be changed to “to understand the possible mechanisms of flax response to salt”. The data presented here does not investigate the mechanisms of salt tolerance because the authors did not compare salt tolerant and salt sensitive flax varieties.

Response 9: Plant salt tolerance mechanism is a complex synthesis. The sentence here listed to illustrate the significance of stress regulation-related differentially expressed genes identified in this study by using high-throughput sequencing technology. We rewrote this part to make our original intention clearly.

Point 10: Some examples of grammatical mistakes

Response 10: We have improved the whole article to eliminate grammatical mistakes. And the title was corrected as: Comprehensive Analysis of Differentially Expressed Unigenes under NaCl Stress in Flax (Linum usitatissimum L.) using RNA-Seq.

Above all, it is very kind of you to give the valuable comments to this manuscript, thank you again!

Yours,

Wu Jianzhong